# Electrochemical Study of Enzymatic Glucose Sensors Biocatalyst: Thermal Degradation after Long-Term Storage

**Marcelinus Christwardana [1,2,*]** and **Domenico Frattini [2]**

[1] Department of Chemical Engineering, Institut Teknologi Indonesia, Jl. Raya Puspiptek Serpong, South Tangerang 15320, Indonesia

[2] Graduate School of Energy and Environment, Seoul National University of Science and Technology, 232 Gongneung-ro, Nowon-gu, Seoul 01811, Korea

* Correspondence: marcelinus@iti.ac.id; Tel.: +62-21-756-1092

**Abstract:** The thermal degradation related to stability in long-term storage of a carbon nanotube-based biosensor has been investigated. The effect of storage temperature on detachment and denaturation of glucose oxidase (GOx) biocatalyst has been proved. The carbon nanotubes (CNTs) coated with polyethyleneimine (PEI) as entrapping polymer to attract more GOx to form a durable and layered CNT/PEI/GOx structure is used for long-term storage to minimize GOx detachment from the structure and minimize the possibility of enzyme and protein denaturation. After 120 days, the glucose response of the CNT/PEI/GOx biosensor stored under 4°C is preserved up to 66.7% of its initial value, while under a 25 °C storage the response is maintained up to 41.7%. The enzyme coverage activity of CNT/PEI/GOx stored at 4 °C and 25 °C has decreased by 31.1% and 51.4%, respectively. Denaturation and detachment of GOx are the common causes of thermal degradation in biosensors under improper storage temperatures, but the presence of PEI in the structure can slow-down these phenomena. Moreover, the electrons transfer constant of CNT/PEI/GOx biocatalyst stored at 4 °C and 25 °C were $7.5 \pm 0.5 \, s^{-1}$ and $6.6 \pm 0.3 \, s^{-1}$, respectively, indicating that also electrons mobility is damaged by detachment and denaturation of enzyme protein and the detection of glucose from the glucose oxidation reaction (GOR) is compromised.

**Keywords:** carbon nanotube; glucose oxidase; glucose sensor; polyethylenimine; thermal degradation

## 1. Introduction

Carbon Nanotubes (CNTs) has long been known as an attractive material in various applications, especially sensors or biosensors due to its unique mechanical and electronic properties [1]. For instance, low bulk densities (1.3 g/cm$^3$), high Young's modulus (more than 1 TPa), high tensile strength (up to 63 GPa), good chemical and environmental stability and high thermal conductivity (~3000 W/m·K), low bandgaps (between 0.18 and 1.8 eV) [2–4]. Beside these advantages, CNTs have the ability to adsorb proteins and bond with them by specific interaction such as covalent bonding (chemical bond formation) and non-covalent bonding (physisorption) [5,6]. CNTs can be modified with an enzyme to get functional biocatalyst structures for biodevices, such as biosensors and biofuel cells.

The enzymes in CNT-based biosensors or biofuel cells for glucose detection and utilization are usually Glucose Oxidase (GOx) or Glucose Dehydrogenase (GDH) as they can convert glucose to protons, electrons, and gluconolactone. Although advanced physical and chemical characterizations about enzyme modified CNTs biocatalysts have been reported by many researchers, the long-term storage stability of the biocatalyst ink and biocatalyst performance is the current issue to be extensively

investigated. Many researchers mentioned that the biocatalyst ink should be stored at low temperature when not used [7–9], to avoid denaturation of proteins inside the enzyme. A deeper characterization about long-term stability, the effect of storage temperature, the decrement of electrochemical properties, and the protection strategies to preserve biocatalyst activity is needed to deeply understand the statement above and suggest protocols to manage these materials for longer times. The effect of storage conditions on biocatalyst ink is important to be examined from the industrial and commercial point of view for the prospective diffusion on large scale. In fact, it is possible to naively use low and very low (i.e., cryogenic) storage temperatures for biocatalysts, but costs will inexorably increase and limiting the application fields. Using higher storage temperatures for a long time will increase the probability of denaturation of proteins and thermal degradation of the biosensor due to the detachment of enzymes from the supporting structure although storage costs are sensibly lower, but stability and durability are not ensured.

Therefore, the relationship between storage time, temperature and electrochemical properties of biocatalyst should be considered as a factor affecting the design, fabrication and production of biosensors industry. In fact, the concept is that a structure with a strongly bonded enzyme, by adding crosslinking components, has higher possibilities to retain its functions over time and at different storage temperatures. Weak bonding between enzymes and supporting CNTs reduces the effectiveness of the structure and does not valorize the use of this expensive nanomaterial because the enzymes can detach from the structure over time and if accidentally exposed to improper temperature thus lifetime is shorter.

The entrapping material considered as supporting material for enzyme immobilization on the surface of CNTs is polyethyleneimine (PEI), which is characterized by many amine groups in its branched structure. PEI has several advantages for biosensors applications, such as low toxicity, ease of separation and recycling, biocompatible, and odorless [10]. PEI possesses a high ionic charge density although being a weak polymeric base with pKa values between 7.9 and 9.6 [11]. Immobilization of enzymes using PEI as entrapping polymer has three main advantages [12]: (i) Enzyme bonding is relatively strong because PEI has multiple cation groups at different distance, (ii) PEI has random structure and enzyme is free to house and arrange on it, and (iii) the polymeric bed formed by PEI offers a three-dimensional protein entrapment which enable to immobilize enzyme up to 80%. Despite its many advantages for enzyme immobilization, the utilization of PEI as entrapping polymer for enzyme immobilization presents some drawbacks, for instance, (i) PEI matrix could interact with some cations in enzyme active sites, (ii) destabilization of enzyme could arise due to PEI mobility, and (iii) sometime, crosslinking stability is relatively poor if branching is not enough, compared to covalent bonding in similar polymers [3,13,14].

In this work, GOx is used as a biocatalyst for glucose biosensing by its Flavin Adenine Dinucleotide (FAD) reaction center which is located inside its hydrophobic pocket. GOx has the ability to oxidize glucose to gluconolactone by using oxygen as an electron acceptor and generates hydrogen peroxide simultaneously. The novelty of the work consists of the detailed study of residual activity and the assessment of storage stability, long term, under two very different temperatures. Electrochemical techniques and related calculations are extensively used to explain and test the benefits of the proposed layered nanostructure involving high conductive CNTs and the PEI wrapping polymer to stabilize enzymes. A glucose biosensor based on CNT, GOx and PEI is investigated and electrochemical performances after 120 days of storage at 4 °C or 25 °C are measured and discussed. The high relative activity, the long term bonding effect of PEI, the simple biocatalyst structure, and durability support the proposed novelty of this work. Two kinds of biocatalyst structures were investigated electrochemically: CNT/GOx (control sample) and CNT/PEI/GOx. The stability is studied by using Cyclic Voltammetry (CV). From CV data, the enzyme coverage activity is measured as well as the electrons transfer rate constant. The response of the biosensor to glucose is used in this work to estimate enzymatic activity. All electrochemical characterizations were measured at the initial day and final day, to quantify the degradation of biocatalyst properties over time.

## 2. Materials and Methods

### 2.1. Materials

The Multiwalled carbon nanotubes (MWCNT, MR99, Average diameter 20 nm, >99%) as a supporting electrode were obtained from Carbon NanoTech (Gyeongbuk, Korea). PEI (50% (*w/v*) solution in water, MW 750,000), GOx (from *Aspergillus niger* type X-S, 100,000-250,000 U/g solid), D-glucose, and Phosphate Buffer Saline (PBS, pH 7.4) were purchased from Sigma Aldrich (Milwaukee, WI, USA).

### 2.2. Biocatalyst Fabrication

CNT/PEI/GOx layers were prepared by the layer-by-layer (LbL) deposition method between GOx on coated MWCNT [15,16]. Initially, 10 mL of 2.5 mg/mL PEI solution (in 0.01 M PBS pH 7.4) and 50 mg of CNT were mixed together, and then sonicated for 10 min and stirred for 2 h. After that, the mixture was centrifuged. Deionized water (DI) was used to remove excess PEI in solution. The mixture was immersed into 5 mg/mL GOx solution for 1 h and then was centrifuged again to remove the supernatant and completing the GOx/PEI/CNT catalyst. The same method was also used to fabricate the CNT/GOx catalyst by immersing CNTs in 5 mg/mL GOx solution. The resulted CNT/GOx is used as a reference sample for measurements. Finally, these two biocatalyst inks were stored at 4 °C and 25 °C. The structure of these catalysts, CNT/GOx and CNT/PEI/GOx are schematized in Figure 1.

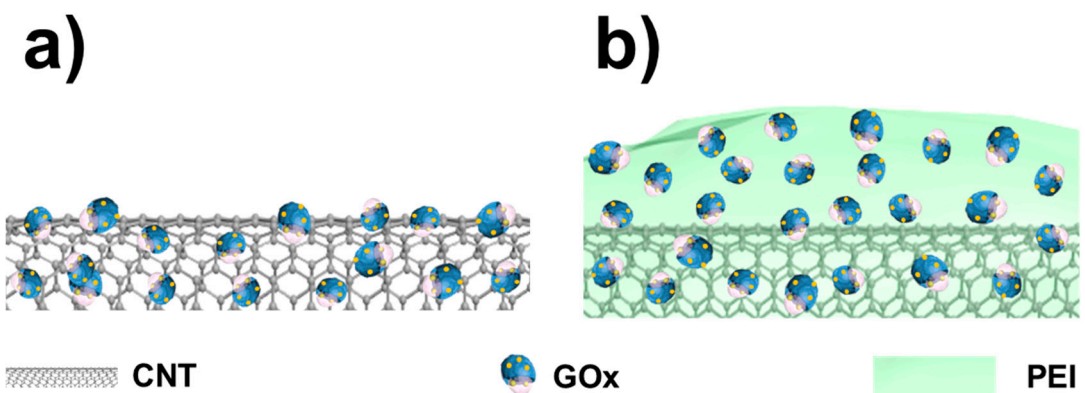

**Figure 1.** Structure of (**a**) carbon nanotube (CNT)/ glucose oxidase (GOx) and (**b**) CNT/PEI (polyethyleneimine)/GOx catalyst.

### 2.3. Electrochemical Measurement

The electrochemical measurements by CV were performed using an SP-240 BioLogic potentiostat (Bio-Logic Science Instruments SAS, Seyssinet-Pariset, France) and data were acquired and processed by using EC-Lab software. In the three-electrode cell measurement, a Pt wire and Ag/AgCl (soaked in 3 M NaCl) were used as a counter and a reference electrode, respectively. A Glassy Carbon Electrode (GCE) loaded with the biocatalyst ink was used as a working electrode (active area 0.194 cm$^2$) [17]. To fix the biocatalyst ink on the surface of GCE, 8 μL of ink was dropped on the GCE surface and air dried. Then, 5 μL of 5 wt% Nafion solution was dropped on the ink-loaded GCE [18] and air dried. PBS 0.1 M was used as electrolyte solution to promote redox reaction in GOx. High purity N$_2$ gas was used to purge the cell and provide anaerobic condition, while environmental air is used for the aerobic condition. The electrochemical characterizations were conducted at the initial time, day-0, and at the final time day-120.

## 3. Results and Discussion

### 3.1. Electrochemical Characterization of Biocatalyst at Day-0

CV curves of CNT/GOx and CNT/PEI/GOx at day-0, under anaerobic condition, are shown in Figure 2a. Accordingly, two phenomena can be described: (i) The redox peaks appeared between $-0.45$ and $-0.50$ V vs. Ag/AgCl, with current peaks centered at potential $(E_p)$ $-0.48$ V vs. Ag/AgCl, which is related to the redox reaction of $FAD/FADH_2$ ($FADH_2 \leftrightarrow FAD + 2e^- + H^+$) on GOx [19]; (ii) the normalized peaks current density (i.e., between background current and peak current) of CNT/PEI/GOx is 0.104 mA·cm$^{-2}$, higher than 0.079 mA·cm$^{-2}$ for CNT/GOx, meaning that, qualitatively, the amount of immobilized GOx in CNT/PEI/GOx was higher than CNT/GOx. This can be explained by the type of bonds in the structures. Although adsorption and hydrophobic interaction are the main bonding between GOx and CNT, they are not strong enough due to the interference of electrostatic charges between them, repelling each other electrostatically because both CNT and GOx have a negative charge at neutral pH [20]. Conversely, in the CNT/PEI/GOx structure, PEI coats CNTs very well and makes the CNT disperse better in DI water because CNT has negative charge and PEI has a positive charge at neutral pH. Therefore, PEI bonded strongly to CNT [20,21] and the charge of CNT/PEI surface is thus positive and GOx, which has a negative charge at neutral pH, is electrostatically attracted by the CNT/PEI surface. This is compatible with the CV results. GOx is attached on the surface of the CNT surface by hydrophobic interaction and a little bit of adsorption, but leakage over time is more prone to happen when the bonding between them is not strong.

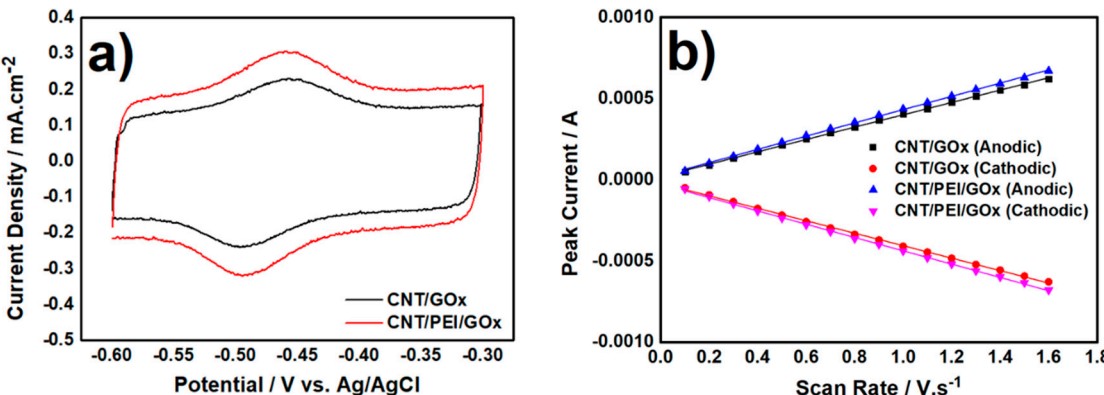

**Figure 2.** (**a**) CV curves of CNT/GOx and CNT/PEI/GOx at scan rate 100 mV·s$^{-1}$. Linear trend between peak current vs. scan rate ranging from 100 to 1600 mV·s$^{-1}$ (**b**). All experiments were conducted under anaerobic condition by feeding $N_2$ gas at 100 mL·min$^{-1}$ and 0.1 M Phosphate Buffer Saline (PBS) pH 7.4 acted as the electrolyte.

The electron transfer mechanism and efficiency of CNT/GOx and CNT/PEI/GOx was investigated by CV measurements, at scan rates from 100 to 1600 mV·s$^{-1}$, according to [22]. The obtained current peaks were plotted vs. scan rate as shown in Figure 2b. The peak current increased linearly with increasing scan rate, meaning that both CNT/GOx and CNT/PEI/GOx have quasi-reversible and surface reaction-controlled regime [23]. Moreover, surface coverage activity (Γ), which is the enzymatic activity of immobilized GOx on the electrode surface area, can be determined from the slope of data in Figure 2b, following Equation (1) [24]:

$$I_P = n^2 F^2 A \Gamma \nu / 4RT, \tag{1}$$

where $I_P$ is peak current, $\nu$ is scan rate, n is the number of the electron involved in the redox reaction, F is Faraday constant, A is electrode area, R is ideal gas constant, and T is temperature. The value of surface coverage activity for CNT/GOx and CNT/PEI/GOx were

$5.25 \times 10^{-10}$ and $5.62 \times 10^{-10}$ mol·cm$^{-2}$. These values are higher if compared to other GOx-based catalyst structures. For instance, ITO/CzS/GOx has $\Gamma$ value of $2.5–2.8 \times 10^{-12}$ mol·cm$^{-2}$ [25], GOx/Ag@MWCNT-IL-Fe$_3$O$_4$/MGCE structure has $\Gamma$ value of $5.84 \times 10^{-11}$ mol·cm$^{-2}$ [26], GOx/OG-CNT/GCE has $1.8 \times 10^{-10}$ mol·cm$^{-2}$ [27], and theoretical GOx monolayer on GCE has $2.86 \times 10^{-12}$ mol·cm$^{-2}$ [28].

For deeper electrochemical characterization, the value of the electron transfer rate constant ($k_s$) can be determined by plotting the redox peak potentials vs. scan rate. The resulted slope between potential and log scan rate was used to estimate $\alpha n$ and $k_s$ following Equation (2) [29]:

$$E_p = E^{0\prime} + (RT/\alpha nF) \, [\log (RT \, k_s/\alpha nF) - \ln v], \tag{2}$$

where $\alpha$ is electron transfer coefficient, $k_s$ is electron transfer rate constant, $E_p$ is peak potential, and $E^{0\prime}$ is formal potential. The results of enzyme coverage activity and electron transfer rate calculations are shown in Table 1.

**Table 1.** Surface coverage activity and electron transfer rate constants of CNT/GOx and CNT/PEI/GOx at day-0.

| Catalyst Structure | $\overrightarrow{\Gamma}^*$ ($\times 10^{-10}$) (mol·cm$^{-2}$) | $\overleftarrow{\Gamma}$ ($\times 10^{-10}$) (mol·cm$^{-2}$) | $\Gamma_{average}$ ($\times 10^{-10}$) (mol·cm$^{-2}$) | $k_s$ (s$^{-1}$) |
|---|---|---|---|---|
| CNT/GOx | 5.23 | 5.30 | 5.25 | $8.5 \pm 0.2$ |
| CNT/PEI/GOx | 5.61 | 5.64 | 5.62 | $10.1 \pm 0.4$ |

* Arrows '$\rightarrow$' and '$\leftarrow$' indicate the oxidation (anodic) and reduction (cathodic) reactions, respectively.

The as-prepared CNT/PEI/GOx structure has more immobilized GOx, thus having higher activity and faster electron transfer rate than the simple CNT/GOx structure. This is clearly ascribed to PEI because the main role is to entrap GOx on the CNT surface, but its effect on durability under storage should be determined too.

GOx based biosensors need dioxygen as a mediator to do glucose oxidation reaction (GOR). For that, CV was also measured under aerobic conditions using environmental air (without additional air/O$_2$ gas) as shown in Figure 3a,b.

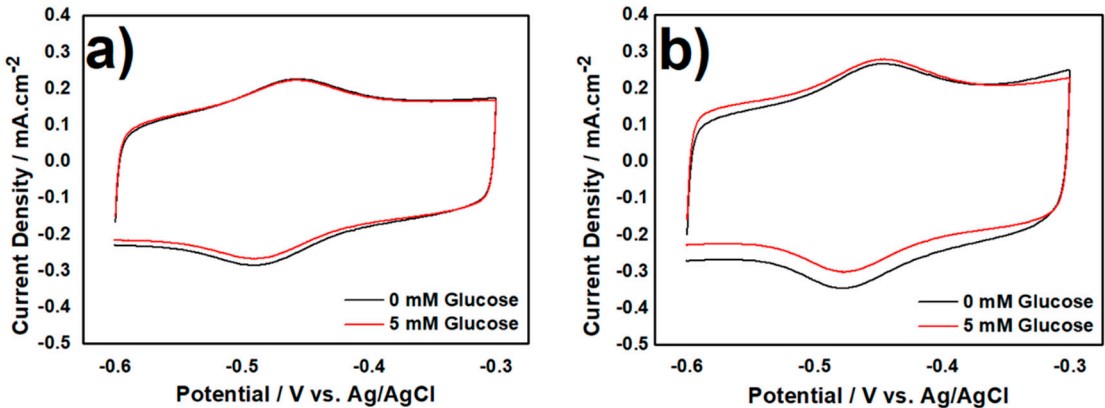

**Figure 3.** CV curves of (**a**) CNT/GOx and (**b**) CNT/PEI/GOx at scan rate 100 mV·s$^{-1}$ with absence (black) and presence (red) of 5 mM glucose in the electrolyte. All experiment conducted under aerobic condition and 0.1 M PBS pH 7.4 acted as the electrolyte.

Compared to CV curves in Figure 2a, the background current of CVs in this condition was shifted downward. This shifted is related to oxygen reduction reaction (ORR) (O$_2$ + 4H$^+$ + 4e$^-$ $\rightarrow$ 2H$_2$O) by CNT [30]. In ORR, the protons and electrons were taken from two sources: Electrolyte solution and FADH$_2$ oxidation reaction (FADH$_2$ $\rightarrow$ FAD + 2H$^+$ + 2e$^-$). When 5 mM glucose was added to

the system, GOR occurred following: Glucose $\rightarrow$ gluconolactone + $2H^+$ + $2e^-$ [31]. In this condition, protons and electrons from GOR were used for the reduction reaction of FAD (FAD + $2H^+$ + $2e^-$ $\rightarrow$ $FADH_2$) and made the background current shifted upward. At the same time, GOx converted the oxygen from glucose to $H_2O_2$ ($O_2$ + $2H^+$ + $2e^-$ $\rightarrow$ $H_2O_2$) [32]. The changes in current, due to these reactions, are usually the signals detected by glucose biosensors. The FAD/$FADH_2$ reduction peak current density for CNT/GOx and CNT/PEI/GOx shifted upward to 26 and 60 $\mu A \cdot cm^{-2}$, respectively.

Again, CNT/PEI/GOx can promote electron transfer better than CNT/GOx. Differently from previous literature, the enzyme-entrapped polymer matrix on CNT uses only PEI as stabilizer, thus avoiding the costly use of polyelectrolyte solutions for impregnation [33]. Moreover, as suggested in Appleton et al. [33], increasing the amount of the stabilizer in the bath solution does not confer further stability or immobilization, hence the immobilization is limited by the PEI uptake of CNT.

### 3.2. Electrochemical Characterization of Biocatalyst at Day-120

#### 3.2.1. CV measurement in Anaerobic Condition

The effect of two storage temperatures (4 °C and 25 °C) on the activity of GOx in catalytic structures, for durability study, was investigated and quantified by CV measurements under anaerobic condition, as shown in Figure 4a,b. The FAD/$FADH_2$ redox peaks from CNT/GOx and CNT/PEI/GOx stored at room temperature (25 °C) are lower, compared to biocatalysts stored at a cool temperature (4 °C). The decreasing in peaks current density is related to two factors: (i) Denaturation of proteins in the active site of GOx and (ii) strength of bonds in biocatalyst during the storage process. The peak current density of CNT/GOx stored at 4 °C and 25 °C decreased by 19.09% and 26.22%, respectively, and peak current density of CNT/PEI/GOx decreased by 6.76% and 13.93%, respectively, from the initial value.

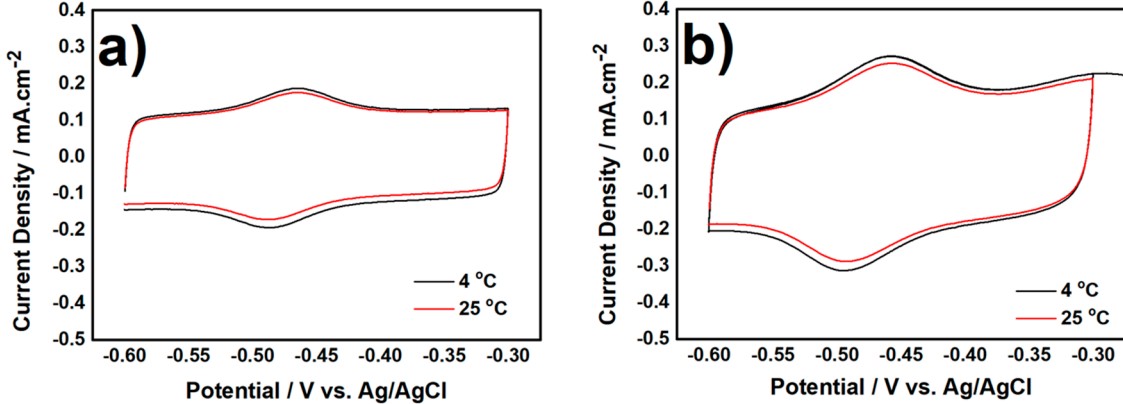

**Figure 4.** CV curves of (**a**) CNT/GOx and (**b**) CNT/PEI/GOx at scan rate 100 $mV \cdot s^{-1}$ stored for 120 days. All experiment under anaerobic condition by feeding $N_2$ gas with flow rate 100 $mL \cdot min^{-1}$ in PBS 0.1 M pH 7.4 as the electrolyte.

Moreover, surface coverage activity and electron transfer rate constant of both biocatalysts were investigated by measuring the peak current at scan rates from 100 to 1600 $mV \cdot s^{-1}$. Based on the plot in Figure 5a,b, the current of both biocatalyst structures, stored at 4 °C and 25 °C, increased linearly as a function of the scan rate. After 120 storage days, the rate determining step is still the surface reaction, i.e., there is no change in mechanism regime, and the chemosensing ability is preserved.

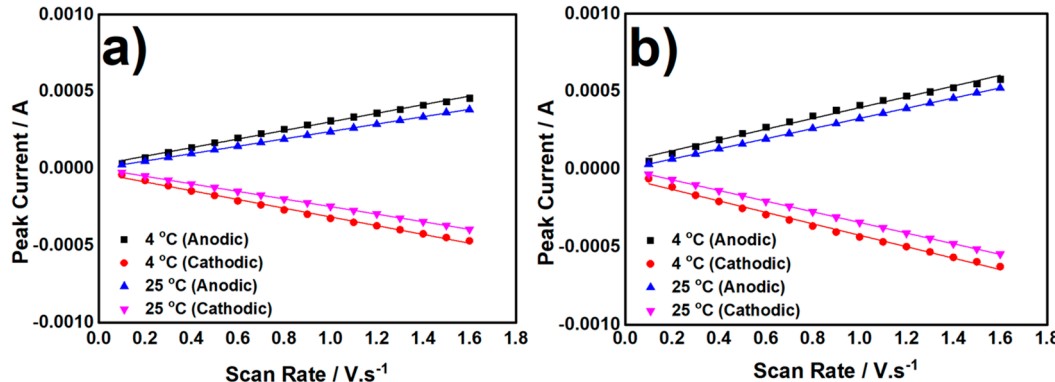

**Figure 5.** Linear line between peak current vs. scan rate at various scan rate from 100 to 1600 mV·s$^{-1}$ of (**a**) CNT/GOx and (**b**) CNT/PEI/GOx which stored at 4 °C and 25 °C for 120 days. All experiment conducted under anaerobic condition by feeding N$_2$ gas with flow rate 100 mL·min$^{-1}$ and 0.1 M PBS pH 7.4 acted as the electrolyte.

The values of surface coverage activity and electron transfer rate constant are shown in Table 2.

**Table 2.** Surface coverage activity and electron transfer rate constants of CNT/GOx and CNT/PEI/GOx at day-120.

| Catalyst Structure | Storage Condition (°C) | $\overrightarrow{\Gamma}$* $(\times 10^{-10})$ (mol·cm$^{-2}$) | $\overleftarrow{\Gamma}$ $(\times 10^{-10})$ (mol·cm$^{-2}$) | $\Gamma_{average}$ $(\times 10^{-10})$ (mol·cm$^{-2}$) | $k_s$ (s$^{-1}$) |
|---|---|---|---|---|---|
| CNT/GOx | 4 | 3.84 | 3.91 | 3.87 | 5.1 ± 0.2 |
| | 25 | 2.69 | 2.76 | 2.73 | 2.8 ± 0.4 |
| CNT/PEI/GOx | 4 | 4.76 | 5.10 | 4.91 | 7.5 ± 0.5 |
| | 25 | 3.66 | 3.84 | 3.75 | 6.6 ± 0.3 |

* Arrows '→' and '←' indicate the oxidation (anodic) and reduction (cathodic) reactions, respectively.

The surface coverage activity, calculated from Equation (1), of CNT/GOx stored at 4 °C and 25 °C for 120 days decreased by 26.29% and 48%, respectively from its initial value (day-0), while in CNT/PEI/GOx decreased by 12.63% and 33.27% when stored at 4 °C and 25 °C, respectively, for 120 days. The decreasing surface coverage activity at long term cannot be completely avoided, and these data confirm that GOx is detached more from the CNT surface if PEI is not adopted. The detachment of GOx from the surface of CNT depends on the bonding between GOx and the surface of CNT and clearly the bonding with CNT/PEI was stronger. The higher storage temperature also affects GOx detachment process. The presence of PEI as entrapping polymer can cut by half the detachment of GOx from the surface of CNT. After long term storage, the CNT/GOx stored at 4 °C and 25 °C for 120 days has $k_s$ value of 5.1 ± 0.2 s$^{-1}$ and 2.8 ± 0.4 s$^{-1}$, respectively and the CNT/PEI/GOx has $k_s$ value of 7.5 ± 0.5 and 6.6 ± 0.3 s$^{-1}$. As explained before, the decreasing electron transfer also indicates the unavoidable degradation of the enzymatic catalyst in long term situations. However, by applying PEI on the CNT surface, GOx detachment can be minimized, so a still high $k_s$ can be maintained. Far lower storage temperatures can be suggested to minimize more the degradation of electrochemical activity, but long term storage below 0 °C is expensive, and the possibility to reduce degradation by preparing suitable catalytic structures, in which GOx is better bonded, can provide more stability in an easy and cost-effective way and is more viable than an obvious very low storage temperature.

### 3.2.2. CV Measurement in Aerobic Condition

Deeper characterization regarding protein denaturation during the storage process was evaluated by measuring CV of biocatalysts with the presence of glucose under the aerobic condition, as shown

in Figure 6. In that condition, glucose can be oxidized to gluconolactone, protons, and electrons as explained before.

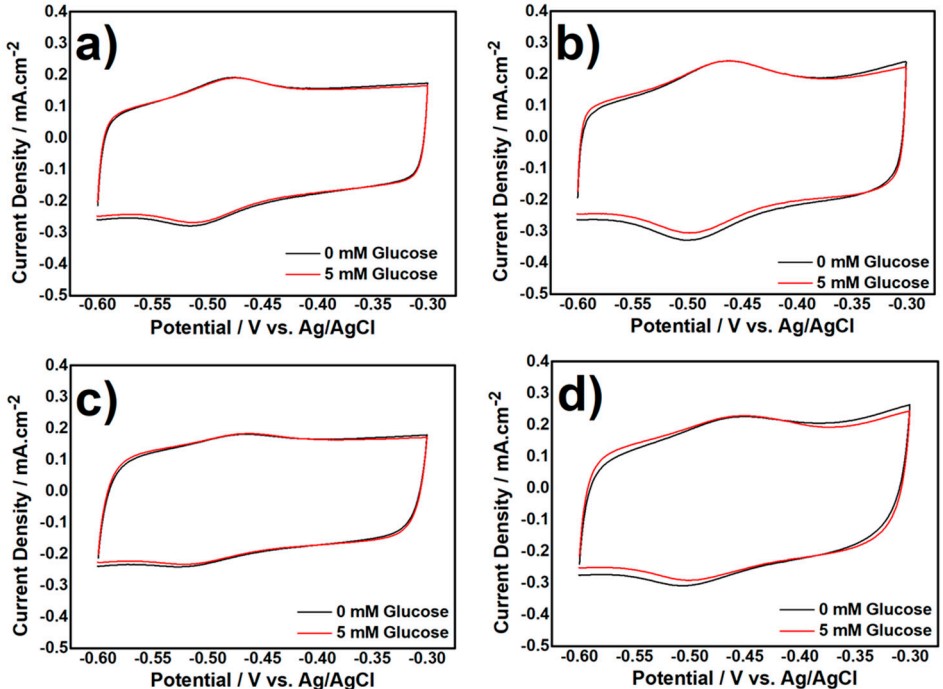

**Figure 6.** CV curves of (**a**) CNT/GOx and (**b**) CNT/PEI/GOx stored at 4 °C; CV curves of (**c**) CNT/GOx and (**d**) CNT/PEI/GOx stored at 25 °C for 120 days. Scan rate is 100 mV·s$^{-1}$ in the absence (black) and presence (red) of 5 mM glucose in the electrolyte. All experiment conducted under aerobic condition in 0.1 M PBS pH 7.4 as the electrolyte.

Several phenomena can be noticed: When glucose is added to the electrolyte, the FAD/FADH reduction peak of CNT/GOx and CNT/PEI/GOx stored at 4 °C shifted up by 11 µA·cm$^{-2}$ and 40 µA·cm$^{-2}$, respectively (Figure 6a,b). The same peak in CNT/GOx and CNT/PEI/GOx stored at 25 °C shifted up by 8 µA·cm$^{-2}$ and 25 µA·cm$^{-2}$, respectively (Figure 6c,d). The upshifting of biocatalyst structures stored at 4 °C were higher compared to those stored at room 25 °C due to denaturation of protein in the active site of GOx, and even this tiny and apparently negligible shift can affect the electron transfer. The capability of GOR from CNT/PEI/GOx to produce electricity was higher than CNT/GOx, even when stored at 25 °C for long-time.

It proves that PEI as entrapping polymer has an important role to maintain GOx activity and preserve it from detachment in the biocatalyst structure. Considering the related literature, the comparison of storage stability between this work and other proposed biocatalyst structures are shown in Table 3.

Biocatalyst storage stability in this work, based on GOR process, is comparable and reasonable considering the simple structure proposed and the long term storage time checked. In fact, not many works investigated a storage time >20–30 days, without using expensive metal nanoparticles like Au and Pt, and information about the long-term stability are few. Therefore, the simple structure, the better stability under two different temperatures, the absence of precious metals and the long term storage possibility are the major benefits claimed for the CNT/PEI/GOx structure proposed here.

**Table 3.** Comparison of long-term storage stability from several glucose biosensor catalysts [1].

| Catalyst Structure | Maintained Storage Stability [2] (%) | Storage Time (Days) | Reference |
|---|---|---|---|
| GOx/Ag@MWCNT-IL-Fe$_3$O$_4$/MGCE | 79.0 | 28 | [34] |
| GOx-NiO | 95.0 | 10 | [35] |
| GOD/GNp/MWCNTs/Pt | 92.0 | 30 | [36] |
| GNPs/Pb NWs | 80.7 | 70 | [37] |
| GOD-chitosan-AuNP-PB/GCE | 70.0 | 14 | [38] |
| PDDA/GOx/PDDA/CNT/GC | 90.0 | 28 | [39] |
| CNT/PEI/GOx | 66.7 | 120 | *This work* |

[1] Based on glucose response method with various glucose concentrations. [2] Maintained percentages from initial value under storage condition at 4 °C.

Moreover, another novel implication, based on the stability results described in this work, is the use of PEI as a single stabilizer component for enzyme-entrapped biosensors. The introduction of PEI in the formulation of enzyme-loaded pellets for re-usable glucose sensors [40] could be helpful to semplify the recipes toward the penetration of on-demand and on-site self-prepared biosensing strips [41] or other graphite-based materials for bioelectrochemical devices [42,43].

## 4. Conclusions

We successfully investigated the long-term stability and activity of CNT/PEI-based GOx biocatalyst nanostructure when stored at cool and room temperature for 120 days. The GOx attached directly on CNT surface is easy to detach from the structure during long term, while GOx attached on CNT/PEI is less easy to be detached during long-term storage. The presence of PEI on the surface of CNT attracted the GOx and firmly attached it on the CNT surface due to the different superficial charge between CNT/PEI and GOx.

The best structure in this work, CNT/PEI/GOx, has enzyme coverage activity around $5.62 \times 10^{-10}$ mol·cm$^{-2}$, electron transfer rate constant of $10.1 \pm 0.4$ s$^{-1}$, and glucose response (5 mM) of 60 µA·cm$^{-2}$ at the initial time. After 120 days of storage under 4 °C, the enzyme coverage activity decreased and was $4.91 \times 10^{-10}$ mol·cm$^{-2}$; $k_s$ value was $7.5 \pm 0.5$ s$^{-1}$, and glucose response is still 45 µA·cm$^{-2}$. On the other side, when CNT/PEI/GOx is stored for 120 days under room temperature, the enzyme coverage activity decreased to $3.75 \times 10^{-10}$ mol·cm$^{-2}$, $k_s$ is $6.6 \pm 0.3$ s$^{-1}$, and glucose response is 25 µA·cm$^{-2}$. Beside maintaining the GOx in the biocatalyst structure, the presence of PEI can slow-down the denaturation of enzyme and natural degradation during the storage process. The relatively high storage stability is also maintained by the adoption of PEI after storage for long-time under room temperature. This high stability allowed to investigate a storage time of 120 days, far longer than other works in the literature for GOx-based biosensors. This study is very useful for biosensor industry to reduce the costs related to biocatalyst storage. The kinetic study about denaturation mechanism of enzyme proteins needs to be further investigated to enhance more stability and to design new structures for GOx.

**Author Contributions:** M.C. and D.F. designed the experiments; M.C. conducted the experiment; M.C. and D.F. analyzed the data; M.C. prepared the original draft of the manuscript; D.F. reviewed and edited the original draft of the manuscript. All authors have given approval to the final version of the manuscript.

**Funding:** This research received no external funding.

**Acknowledgments:** Authors would like to thanks to Energy and Environmental Material and Process (EEMP) Laboratory, Seoul National University of Science and Technology for their facilities.

**Conflicts of Interest:** The authors declare no conflict of interest.

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
