# Peer review of "Electrochemical Study of Enzymatic Glucose Sensors Biocatalyst: Thermal Degradation after Long-Term Storage"

_chemosensors, doi:10.3390/chemosensors6040053_

Round 1
Reviewer 1 Report
Christwardana and Frattini describe a work related to the importance of PEI for stabilizing glucose sensors. Overall the work is good and should be of interest to the readers of the journal. I have one minor comment:
Please compare the performance with examples that include enzyme stabilizers for improved stability. For example:
a. Appleton et al. Sensors and Actuators B 43 (1997) 65 – 69
b. Bandodkar et al. Biosensors and Bioelectronics 101, 181-187
c. Bandodkar et al. Advanced healthcare materials 4 (8), 1215-1224
Author Response
The Authors would like to thank the reviewer for his/her general appreciation of the work. We addressed the minor revisions suggested as follow. We appreciated this comment to improve the comparison with literature. The suggested examples were included in the work and comparison was done appropriate with a brief discussion.
Reviewer 2 Report
Authors report the work "Electrochemical Study of Enzymatic Glucose Sensors Biocatalyst: Thermal Degradation After Long-term Storage". It is important issue for developing long-term storage sensor and they provide systemic study for the work that I thought it could be accepted for publishing in Chemosensors after minor revised.
1. The novelty should be added to the main text.
Author Response
The Authors would like to thank the reviewer for his/her general appreciation of the work. We addressed the minor revisions suggested as follow. We thank the reviewer because we were not sufficiently precise in the main text. A critical discussion of the novelty of the work was added and expanded in the main text.
"The novelty of the work consists in the detailed study of residual activity and the assessment of storage stability at long term, under two very different temperatures. Electrochemical techniques and related calculations are extensively used to explain and test the benefits of the proposed layered nanostructure involving high conductive CNTs and the PEI wrapping polymer to stabilize enzymes. A glucose biosensor based on CNT, GOx and PEI is investigated and electrochemical performances after 120 days of storage at 4°C or 25°C are measured and discussed. The high relative activity, the long term bonding effect of PEI, the simple biocatalyst structure, and durability support the proposed novelty of this work."